# *Mycoplasma Pneumoniae* bronchiolitis and hypoxemia: A retrospective cohort study on risk and prognosis

Yu Chen[ID][1]*, ChenXi Lin[2], Rui Huang[1], Qi Chen[1], Min Zhang[1], XingQian Lai[1], QiaoRu Lin[1], Ling Chen[1]

**1** Department of Pediatrics, Zhongshan Hospital, Xiamen University, Xiamen, Fujian, China, **2** Department of Pediatrics, Shanghai Pudong Hospital, Fudan University, Pudong Medical Center, Shanghai, China

\* hhhanleng@xmu.edu.cn

## Abstract

### Objective

*Mycoplasma pneumoniae* (MP) bronchiolitis can potentially lead to severe respiratory symptoms and long-term complications. This study aimed to determine the risk factors for the development of hypoxemia in MP bronchiolitis and report its prognosis.

### Methods

From January 2017 to December 2024, a total of 178 children with MP bronchiolitis, including 53 cases in the hypoxemia group and 125 cases in the control group, were selected. The clinical data, laboratory indicators, and imaging findings of the two groups were compared. Binary logistic regression analysis was used to identify the risk factors for the development of hypoxemia, and the receiver operating characteristic curve was employed to validate the predictive effect of the risk factors on hypoxemia.

### Results

The hypoxemia group exhibited a higher incidence of a history of allergic diseases and wheezing sounds, accompanied by substantial elevations in C-reactive protein levels and greater areas of CT involvement ($P < 0.05$). The presence of a history of allergic diseases, wheezing sounds, and the number of infected lung lobes were independent risk factors for the development of hypoxemia. The group with hypoxemia demonstrated a delayed improvement in symptoms, signs and lung function during follow-up ($P < 0.05$). Seven cases of bronchiolitis obliterans were diagnosed in the hypoxemia group while none in the control group.

### Conclusion

MP bronchiolitis patients with a history of allergic diseases, wheezing sounds, and involvement of at least three lung lobes are prone to developing hypoxemia. And

**Data availability statement:** All relevant data are within the paper and its Supporting Information files.

**Funding:** The author(s) received no specific funding for this work.

**Competing interests:** The authors have declared that no competing interests exist.

those who experience hypoxemia recover more slowly during short-term follow-up and have a higher incidence of bronchiolitis obliterans.

## Introduction

*Mycoplasma pneumoniae* (MP) is a common respiratory pathogen in childhood with a variety of clinical symptoms. While most MP infections show alveolar consolidation on lung imaging, a less common manifestation is bronchiolitis [1]. Currently, there are limited researches on MP bronchiolitis. It often presents with a high fever and cough, similar to *Mycoplasma pneumoniae* pneumonia (MPP); however, wheezing occurs significantly more frequently [2], and potential respiratory failure has also been noted [3]. Furthermore, patients with the sequelae of bronchiolitis obliterans (BO) have more imaging findings of bronchiolitis during the acute phase [4]. Therefore, pediatricians should pay more attention to MP bronchiolitis due to the potential for severe cases and the likelihood of a worse prognosis.

The risk factors for severe bronchiolitis are listed in widely accepted guidelines, which include aspects such as age in months, history of preterm birth, underlying disorders, and immunological status [5–7]. Notably, these guidelines are primarily focused on the infant population, with background researches concentrating on viral etiologies. MP bronchiolitis differs significantly from classic infant bronchiolitis in terms of major manifestations such as age of onset and fever pattern [1], suggesting that the applicability of existing guidelines may be limited in MP bronchiolitis.

Some studies have identified predictors of severe MP infection [8,9]. However, these studies didn't address the specific imaging patterns of bronchiolitis. Given that imaging differences may reflect different pathological processes leading to different clinical features [10], further studies for the risk factors of MP bronchiolitis are required.

In light of these, the main objective of our study is to identify the risk factors for severe MP bronchiolitis. Since hypoxemia is prevalent in severe cases, it can be regarded as an indicator of severity. Moreover, our study also presents the patients' follow-up status in order to improve the understanding of the prognosis.

## Materials and methods

### Patients

The flowchart for cohort selection procedures is provided in Fig 1. From January 2017 to December 2024, our hospital admitted a total of 16,227 cases of lower respiratory tract infections, among which 4,538 cases were confirmed to have MP infection through paired serum antibody testing and/or nucleic acid testing of secretions. Subsequently, these cases were classified according to the dominant imaging type, and 269 cases of MP bronchiolitis were identified. The main findings on lung CT scans were tree-in-bud sign, centrilobular nodules, and bronchiolar wall thickening. These children constituted the entire cohort of MP bronchiolitis patients treated at this tertiary hospital during that period, and the number of these cases was the sample size for this study. Standardized care was provided in accordance with the Chinese expert consensus on the diagnosis and treatment of MP pneumonia in children (2015 edition). In detail, macrolides were

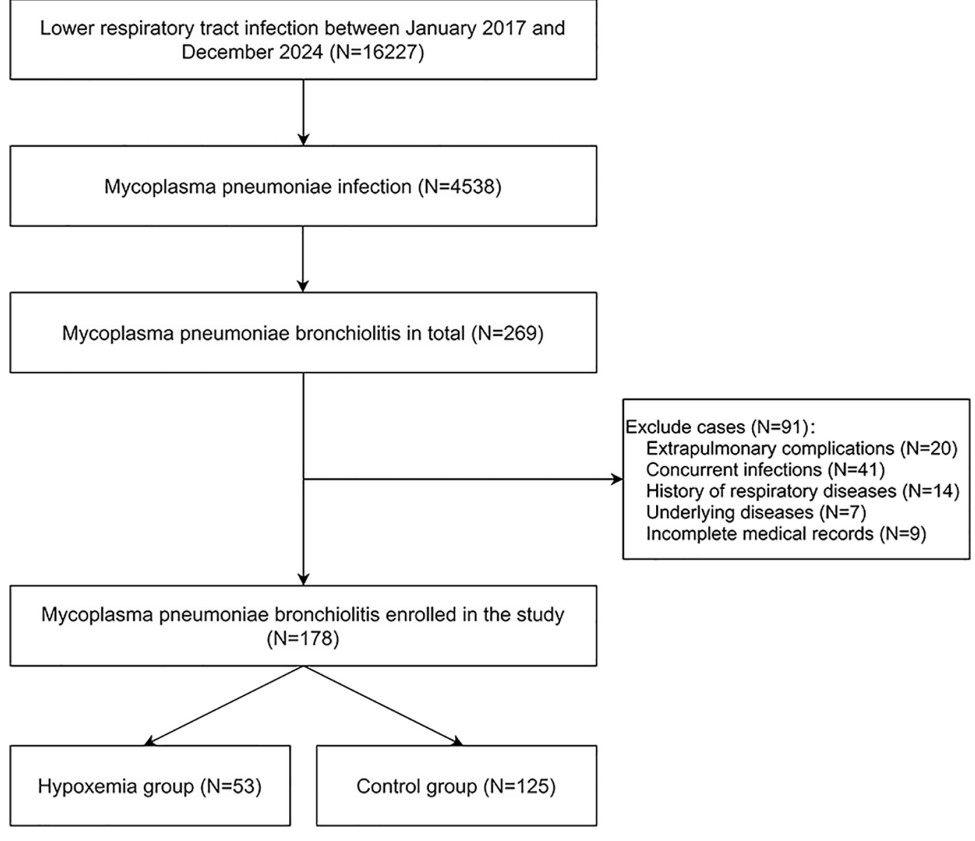

**Fig 1. Flowchart for cohort selection procedures.**

the preferred antibiotics. However, for patients with macrolide resistance mutation who failed to show temperature improvement after 5–7 days of treatment, the antibiotic therapy was switched to doxycycline. For those with acute onset, rapid and severe progression, routine doses of corticosteroids, usually methylprednisolone at 1–2 mg/kg/day were administered for 3–5 days. And pulse doses of corticosteroids were considered when routine doses were ineffective.

Exclusion criteria included: Extrapulmonary complications; concurrent bacterial, viral, tuberculosis, or fungal infections; history of respiratory diseases such as bronchopulmonary dysplasia, airway malformations, or cystic fibrosis; underlying diseases such as congenital heart disease, autoimmune diseases, tumors, etc.; and incomplete medical records due to treatment discontinuation or loss to follow-up for patient-related reasons. Based on these criteria, 20, 41, 14, 7, and 9 children were excluded respectively, leaving 178 children in the study.

Hypoxemia is typically defined as peripheral oxygen saturation <90% or arterial blood gas analysis with $PaO_2 < 60$ mmHg [11], reflecting the need for oxygen therapy. The percutaneous blood oxygen saturation was continuously monitored throughout the entire diagnostic and treatment process. Blood gas analysis was conducted upon admission and in situations when patients experienced respiratory distress. The patients were divided into the hypoxemia group and the control group depending on the presence or absence of hypoxemia.

## Data acquisition and parameter definition

Data were collected by accessing both inpatient and outpatient medical record systems, coupled with telephone follow-up. The demographic and clinical data collected for each patient recruited included the gender, age, year of onset, season of

onset, history of allergic diseases, duration and severity of fever prior to admission, length of cough prior to admission, signs (wet rales, wheezing sounds), time from symptom onset to treatment initiation, and therapies before hypoxemia occurred including the use of antibiotics and steroids. Venous blood samples were collected from all children within 24 hours of admission, and laboratory indicators such as white blood cell count, C-reactive protein (CRP), lactate dehydrogenase (LDH), ferritin, D-dimer, allergen-specific IgE and macrolide resistance mutation were measured by the same operator in the same laboratory. Thoracic CT scans were obtained within three days before or after admission. Due to the prevalence of bronchiolar lesions in all participating patients, a double-personnel counting was resorted to for counting the number of lung lobes infected for imaging comparisons. During the follow-up visit after discharge, short-term prognosis represented by the recovery time of symptoms and signs, as well as the recovery time of lung function was recorded. Cases with long-term complications were excluded due to the difficulty in accurately defining the recovery time. Lung function assessments were performed on all patients after every 14 days to assess the recovery of their small airway injury. Lung imaging was performed to re-evaluate patients with persistently abnormal lung function or symptoms persisting for longer than six weeks to rule out BO. The diagnosis of BO was based on the following criteria [12]: (1) persistent or recurrent wheezing, cough, dyspnea, exercise intolerance, wet rales, and wheezing that persist for more than six weeks following an MP infection; (2) obstructive or mixed ventilatory dysfunction on pulmonary function testing; (3) characteristic radiological findings including mosaic perfusion, air trapping, bronchial wall thickening, and bronchiectasis; and (4) exclusion of other chronic lung diseases.

## Statistical analysis

SPSS 19.0 and GraphPad Prism 10 were used to perform all statistical analyses. The two-sample t-test was employed for comparing two groups of normally distributed continuous variables, with data presented as mean ± standard deviation ($\bar{x} \pm s$). The nonnormally distributed continuous variables were described using the median (interquartile range) [M (P25, P75)], and comparison between groups was done through the Mann–Whitney U test. Percentages or composition ratios (%) were used to represent categorical variables, and the $\chi^2$ test or Fisher's exact probability method was used for comparing two groups. Spearman correlation analysis was adopted to assess the correlation between relevant variables. Variables with a P-value < 0.2 were sequentially subjected to univariate binary logistic regression analysis for variable selection and linear regression for collinearity exclusion, after which multivariate binary logistic regression was performed to determine the independent risk factors associated with hypoxemia, and the model's goodness-of-fit was assessed using the Hosmer-Lemeshow test. The predictive performance of significant continuous variables was evaluated using receiver operating characteristic (ROC) curves. The area under the curve and its 95% confidence interval were calculated. The optimal cut-off value, sensitivity, and specificity were determined based on the maximum Youden's index. P < 0.05 was considered statistically significant.

## Ethics approval and consent to participate

The study was conducted in accordance with the Declaration of Helsinki and approved by the Ethics Committee of Zhongshan hospital, Xiamen University (XMZSYYKY IRB 2024−048). The data were anonymously provided by the Information Department and the collection was completed on May 14, 2025. Due to its retrospective nature and anonymous processing, informed consent was waived.

## Results

### Comparison of demographic and clinical data

A total of 178 patients with MP bronchiolitis were included in the study. Of these, 164 cases (92.1%) were positive for MP DNA, while 113 cases (63.5%) had positive paired serum tests, 36 cases (20.2%) showed negative serological test results (negative initial sera or positive initial sera without 4-fold convalescent titer change), and 29 cases (16.3%) did

not undergo a recheck of serum antibodies. Among the 178 patients, 53 patients belonged to the hypoxemia group and 125 patients were in the control group. The gender ratio, age, and age distribution did not differ statistically significantly between the hypoxemia group and the control group ($P > 0.05$), as evident from Table 1. Preschool (3–6 years) and school-age (≥7 years) children accounted for the majority population in both groups, which is consistent with the age at which the MP infection is most likely to occur. When categorized based on the year of onset, the annual average case count increased significantly compared to the period before the COVID-19 outbreak, but the incidence of hypoxemia was not observed to increase ($P > 0.05$). There was no association between hypoxemia and the season of onset ($P > 0.05$). Of the 178 patients enrolled, 84 cases (47.2%) displayed a history of allergic diseases, with the difference between the two groups being statistically significant ($P < 0.05$), indicating a potential link between allergic disease history and the development of hypoxemia. Cough symptoms were experienced by all patients. However, 10 patients, one in the hypoxemia group and nine in the control group did not display the symptom of fever. Both groups did not demonstrate significant differences in terms of fever duration, fever severity, and cough duration ($P > 0.05$). Physical examination revealed a higher incidence of wheezing sounds in the hypoxemia group ($P < 0.05$), although the incidence of audible wet rales was not significantly different between the two groups ($P > 0.05$). All children with hypoxemia were given corticosteroid therapy after the onset of hypoxemia due to their severity. However, prior to the onset of hypoxemia, there was no difference between the two groups in terms of timing of treatment, use of antibiotics, or application of steroids ($P > 0.05$).

## Comparison of laboratory indicators

As demonstrated in Table 1, the hypoxemia group had significantly elevated CRP levels compared with the control group ($P < 0.05$). However, no statistically significant differences were observed between the two groups in terms of leukocyte, LDH, ferritin, D-dimer levels, and MP DNA load ($P > 0.05$). After excluding 10 cases in the control group that declined testing, the incidence of macrolide resistance mutation was equal in the two groups ($P > 0.05$). The incidences of aeroallergens and food allergens were similar between the two groups ($P > 0.05$).

## Comparison of imaging data

The CT manifestations of bronchiolitis include a tree-in-bud sign, centrilobular nodules and bronchiolar wall thickening [13]. The pulmonary CT scans of all patients displayed one or more of these imaging abnormalities (Fig 2 A-E). Each case in the hypoxemia group had the involvement of three or more lung lobes and exhibited a considerably higher area of lung infection ($P < 0.05$). After excluding 4 MP-DNA-negative cases in the hypoxemia group and 10 MP-DNA-negative cases in the control group, no correlation was found between MP-DNA load and the number of infected lung lobes (correlation coefficient = −0.052, $P = 0.512$).

## Analysis of risk factors for hypoxemia in MP bronchiolitis

The presence or absence of hypoxemia was used as the dichotomous dependent variable, while the independent variables included a history of allergic diseases, positive aeroallergen test, positive food allergen test, wheezing sounds, CRP, D-dimer, and the number of infected lung lobes. Univariate binary logistic analysis showed that only D-dimer was excluded due to its $P$-value > 0.2, whereas all other variables with a $P$-value < 0.2 were carried forward to subsequent analyses. Collinearity analysis based on linear regression was conducted after the two continuous variables, CRP and the number of infected lung lobes, had been subjected to Z-score standardization. The analysis revealed that the variance inflation factors of these variables were 1.191, 1.211, 1.032, 1.099, 1.302, and 1.348, respectively, indicating no significant multicollinearity. Thus, they were included in the multivariate binary logistic regression model. The results demonstrated that a history of allergic diseases, wheezing sounds, and the number of infected lung lobes were independent risk factors for hypoxemia (Fig 3). The Hosmer-Lemeshow test showed a $P$-value of 0.315, suggesting good goodness-of-fit of the

**Table 1. Comparison of clinical data, laboratory indicators and imaging data.**

| Project | Hypoxemia group(N = 53) | Control group (N = 125) | $\chi^2/t/z$ | P value |
|---|---|---|---|---|
| Gender (Male: female) ▴ | 29:24 | 64:61 | 0.185 | 0.668 |
| Age (year) △ | 6.1±2.5 | 5.9±2.7 | 0.279 | 0.780 |
| Age distribution (case, %) ▴ | | | 1.557 | 0.459 |
| < 3 years | 7 (13.2) | 13 (10.4) | | |
| 3-6 years | 26 (49.1) | 74 (59.2) | | |
| ≥7 years | 20 (37.7) | 38 (30.4) | | |
| Year of onset (case, %) ▴ | | | | |
| 2017-2019 | 8 (15.1) | 21 (16.8) | 0.079 | 0.778 |
| 2020-2024 | 45 (84.9) | 104 (83.2) | | |
| Annual average case count (case) | | | | |
| 2017-2019 | 2.7 | 7.0 | | |
| 2020-2024 | 9.0 | 20.8 | | |
| Season of onset (case, %) ▴ | | | 2.388 | 0.496 |
| Spring | 10 (18.9) | 15 (12.0) | | |
| Summer | 23 (43.4) | 59 (47.2) | | |
| Autumn | 13 (24.5) | 39 (31.2) | | |
| Winter | 7 (13.2) | 12 (9.6) | | |
| Atopic constitution (case, %) | | | | |
| History of allergic diseases ▴ | 37 (69.8) | 47 (37.6) | 15.496 | 0.000 |
| Positive aeroallergen test ▴ | 26 (49.1) | 47 (37.6) | 2.019 | 0.155 |
| Positive food allergen test ▴ | 28 (52.8) | 51 (40.8) | 2.182 | 0.140 |
| Fever duration (day) △ | 4.0±1.4 | 4.1±2.1 | −0.328 | 0.744 |
| Fever severity (case, %) ○ | | | 3.139 | 0.354 |
| High fever (> 39.1°C) | 40 (75.5) | 91 (72.8) | | |
| Moderate fever (38.1–39°C) | 7 (13.2) | 19 (15.2) | | |
| Low fever (37.1–38°C) | 5 (9.4) | 6 (4.8) | | |
| Afebrile | 1 (1.9) | 9 (7.2) | | |
| Cough duration (day) △ | 4.5±1.9 | 4.5±2.4 | 0.009 | 0.992 |
| Wet rales (case, %) ▴ | 18 (34.0) | 45 (36.0) | 0.068 | 0.795 |
| Wheezing sounds (case, %) ▴ | 37 (69.8) | 47 (37.6) | 15.496 | 0.000 |
| Leukocyte (×10⁹/L) △ | 8.9±2.8 | 8.5±2.6 | 0.862 | 0.390 |
| CRP (mg/L) ○ | 13.5 (6.8, 30.1) | 7.7 (4.0, 18.0) | −2.548 | 0.011 |
| LDH (IU/L) ○ | 312.8(255.2,425.2) | 311.0(231.4,361.1) | −1.229 | 0.219 |
| Ferritin (µg/L) ○ | 338.0(205.5,532.0) | 317.0(204.0,424.5) | −0.934 | 0.350 |
| D-dimer (mg/L) ○ | 0.79(0.38,1.18) | 0.58(0.42,0.85) | −1.389 | 0.165 |
| MP DNA load (copy/ml) ○ | 62000 (9200, 255000) | 68000 (9900, 390000) | −0.959 | 0.337 |
| Macrolide resistance mutation (case, %) ▴ | 40 (75.5) | 91 (79.1) | 0.283 | 0.595 |
| Number of infected lung lobes ○ | 4 (3,4) | 2 (2,3) | 7.592 | 0.000 |
| Time from symptom onset to treatment initiation (day) ○ | 4.0 (3.0,4.0) | 4.0 (3.0,5.0) | −0.404 | 0.686 |
| Doxycycline usage(case, %)▴ | 9 (17.0) | 24 (19.2) | 0.121 | 0.728 |
| Routine doses of methylprednisolone (case, %)▴ | 17 (32.1) | 33 (26.4) | 0.594 | 0.441 |
| Pulse doses of corticosteroids (case, %)○ | 0(0) | 0(0) | – | – |

△ Independent Samples T-Test; ▴ Chi-Square Test; ○ Fisher's Exact Test; ◇ Mann-Whitney U Test;

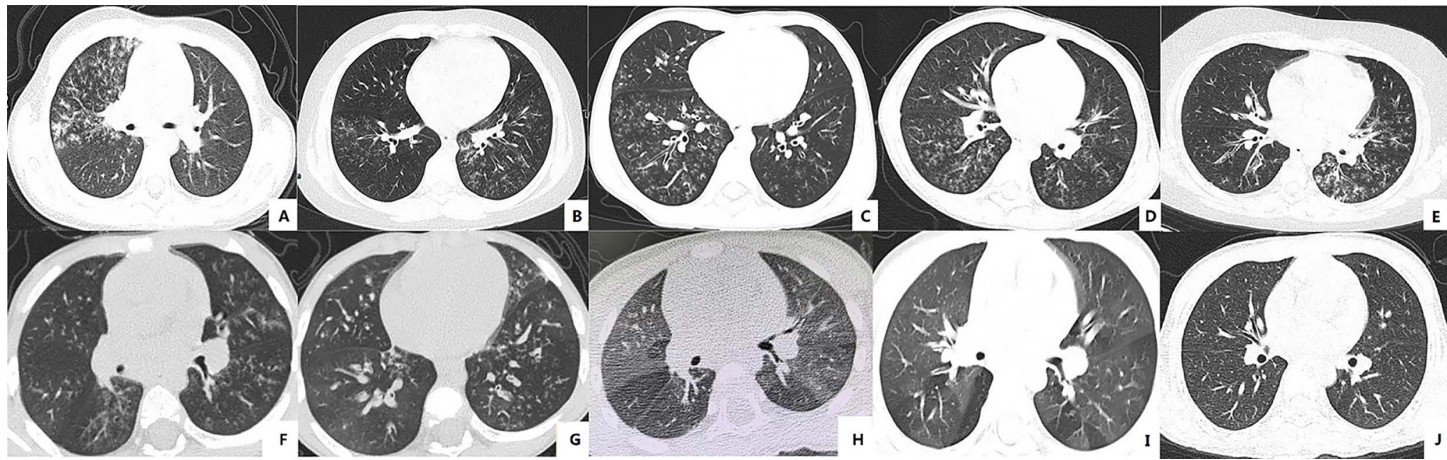

**Fig 2. Representative CT findings of MP bronchiolitis (A-E) and a patient with BO (F-I): lobular central nodule and tree-in-bud sign in all A-E images, tracheal wall thickening in images B, C, and E, and enhanced translucency indicating compensatory ventilation in image D.** Image F and G display lobular central nodules, tree-in-bud opacities, and enhanced translucency in the acute phase, image H shows mosaic perfusion and air trapping two months later, and image I reveals mosaic perfusion and air trapping with partial improvement two years later. Image J is a normal CT image used for comparison.

**univariate binary logistic analysis**

| Variable | OR (95%CI) | P |
|---|---|---|
| History of allergic diseases | 3.710 (1.864-7.384) | 0.000 |
| Positive aeroallergen test | 1.598 (0.835-3.058) | 0.157 |
| Positive food allergen test | 1.625 (0.851-3.102) | 0.141 |
| Wheezing sounds | 3.838 (1.927-7.645) | 0.000 |
| CRP | 1.029 (1.006-1.052) | 0.013 |
| D-dimer | 0.980 (0.841-1.141) | 0.791 |
| Number of infected lung lobes | 5.568 (3.240-9.568) | 0.000 |

**multivariate binary logistic analysis**

| Variable | OR (95%CI) | P |
|---|---|---|
| History of allergic diseases | 2.830 (1.122-7.134) | 0.027 |
| Positive aeroallergen test | 0.522 (0.186-1.463) | 0.216 |
| Positive food allergen test | 1.189 (0.497-2.845) | 0.697 |
| Wheezing sounds | 4.448 (1.739-11.381) | 0.002 |
| CRP | 0.980 (0.946-1.015) | 0.255 |
| Number of infected lung lobes | 6.938 (3.059-13.716) | 0.000 |

**Fig 3. Forest plots displaying risk factors for hypoxemia in MP bronchiolitis: Upper panel: results of univariate binary logistic regression analysis.** Lower panel: independent risk factors for hypoxemia in MP bronchiolitis via multivariate binary logistic regression analysis.

model. In addition, the ROC curve (Fig 4), with an area under curve of 0.843 and a 95% confidence interval of 0.787–0.899, was used to assess the predictive significance of the number of infected lung lobes for hypoxemia. The optimal cutoff value was 2.5 lobes, with a sensitivity of 100% and a specificity of 52.8%. These findings indicate a strong correlation between the number of infected lung lobes and the likelihood of hypoxemia.

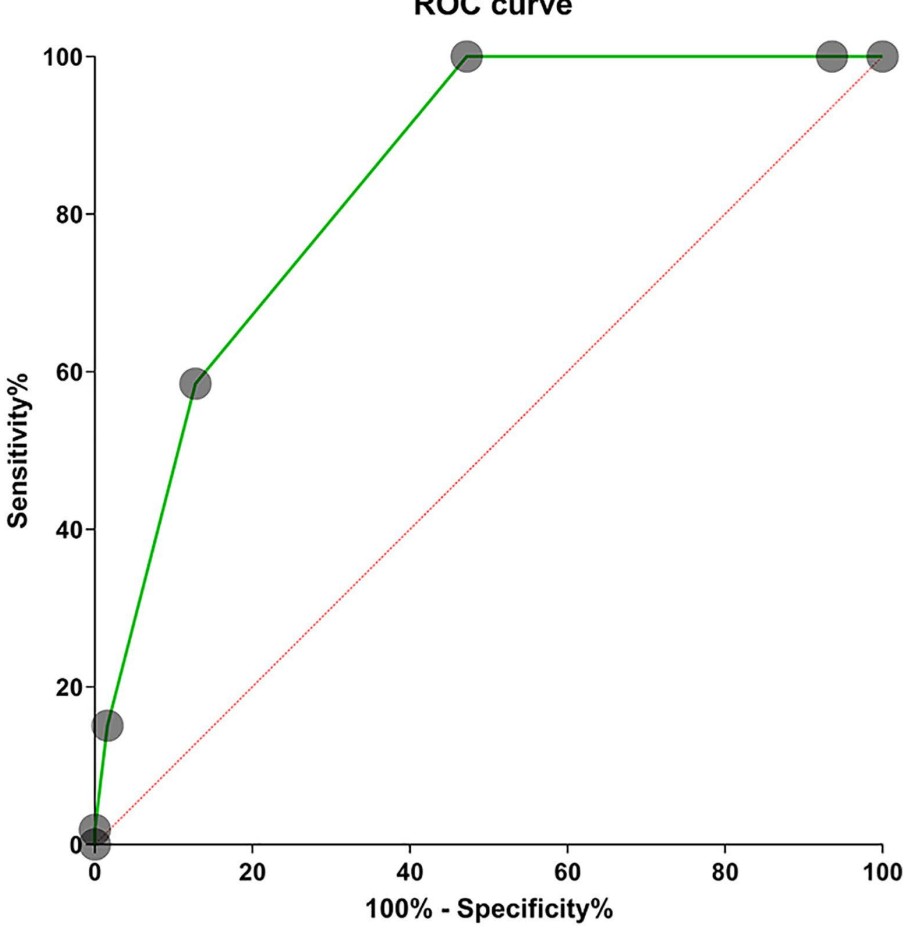

**Fig 4. ROC Curve for predicting hypoxemia based on the number of infected lung lobes.**

## Results of follow-up

The hypoxemia group had a prolonged recovery period from symptoms and signs compared to the control group (18.8 ± 5.3 days vs. 11.7 ± 3.9 days, $t = 9.611$, $P = 0.000$). At the 14-day, 28-day, and 42-day follow-ups, the number of patients with normalized lung function in the hypoxemia group were 15 (32.6%), 21 (45.7%), and 10 (21.7%) respectively, compared to control group, which had 70 (56.0%), 46 (36.8%), and 9 (7.2%) patients, indicating significant differences between the two groups ($x^2 = 10.771$, $P = 0.005$). In the hypoxemia group, there were seven cases of BO, while no such complication was noticed in the control group, indicating a significant difference in incidence ($P = 0.000$). After excluding MP-DNA-negative cases and BO cases, MP-DNA load showed no correlation with the recovery time of symptoms and signs (correlation coefficient = −0.051, $P = 0.524$) or the recovery time of lung function (correlation coefficient = −0.013, $P = 0.870$). S1 Table presents the relevant clinical data, laboratory indicators, and imaging data for patients who developed sequelae of BO during their entire hospitalization period. Pulmonary function exhibited obstructive ventilatory dysfunction, and thoracic CT scans revealed characteristic abnormalities such as mosaic perfusion and gas trapping (Fig 2 F-I).

## Discussion

Our study indicates that a history of allergic diseases, wheezing sounds, and the number of infected lung lobes are independent risk factors for the development of hypoxemia in MP bronchiolitis. Children with hypoxemia recover more slowly during short-term follow-up and have a higher incidence of BO.

Atopic constitutions usually manifest as a history of allergic diseases and positive allergen test results [14]. A few studies have suggested that children with an atopic constitution are more susceptible to bronchiolitis [15], suggesting the importance of considering atopic constitution in predicting the severity of MP bronchiolitis. Dysregulation and overactivation of the immune response are observed following MP infection. The MP-secreted Community-Acquired Respiratory Distress Syndrome (CARDS) toxin facilitates upregulation of cytokines interleukin-4 and interleukin-13, thereby amplifying eosinophilic airway inflammation and promoting a T-helper 2 (Th2) cell response [16]. The pre-existing Th2 – predominant state in individuals with atopic constitutions overlaps with the effects of the CARDS toxin, leading to an amplified and prolonged eosinophilic inflammatory response. Additionally, individuals with an atopic constitution may have a higher MP load in their airways [14,17], accompanied by delayed pathogen clearance, which in turn increases the risk of severe disease [18,19]. Our study indicates that a history of allergic diseases is an independent risk factor for hypoxemia. Therefore, assessing patients' allergic status during the initial evaluation is crucial for the timely identification of early severe MP bronchiolitis.

In our study, wheezing sounds were more frequently observed among those in the hypoxemia group. This could be attributed to the fact that the bronchioles are prone to constriction during exhaling, due to the lack of cartilage support. After infection, mucosal swelling, increased secretions, and shedding of cells can easily lead to airway stenosis. The resulting ventilation impairment can lead to hypoxemia and respiratory failure. Furthermore, the lipoproteins from MP can act as allergens, thereby triggering an IgE-mediated immune response among individuals with atopic constitution [20], and cause symptoms such as airway spasms and wheezing. Thus, it is very important to continuously monitor oxygen saturation in children with MP bronchiolitis when wheezing sounds are auscultated, for promptly identifying and treating hypoxemia.

In cases of diffuse lung disease, severe manifestations can arise due to the depleted normal lung tissue and compensatory ventilation, making the number of involved lung lobes a crucial aspect of our concern. This study demonstrated that the number of infected lung lobes is an independent risk factor for hypoxemia in MP bronchiolitis, particularly when the number exceeds 2.5, indicating a strong predictive value. In clinical practice, if a child has involvement of ≥3 lung lobes, vigilance is recommended and the child must be observed for the occurrence of hypoxemia.

One hypothesis is that diffuse bronchiolar lesions and the occurrence of BO appear to be related, as suggested by Huang *et al*. and Zhao *et al*. [1,4]. In this study, the hypoxemia group had involvement of at least three lung lobes, correlating with an elevated risk of developing BO. Hence, our research supports the hypothesis from a clinical observation perspective. The pathogenesis of BO is a continuous process [21]. During MP infection, the CARDS toxin directly induces necrosis and shedding of airway epithelial cells [16]. Meanwhile, infiltrated neutrophils release proteolytic enzymes and reactive oxygen species, contributing to airway damage [22]. Additionally, CD8 + cytotoxic T lymphocytes target infected host cells, amplifying epithelial injury. These factors collectively exacerbate airway epithelial destruction and exfoliation [23]. Exposure of the damaged basement membrane promotes the formation of reticular exudates by fibrinogen in the bronchiolar lumen, which further initiates the organization process, induces fibroblast infiltration, proliferation, and collagen deposition, and ultimately leads to the formation of irreversible scar tissue, resulting in airway obstruction. Macrophages may also serve a bridging role in this pathological process by regulating inflammatory and fibrotic responses [24,25].

The limitations of this study are the relatively small sample size of single-center, and the possible selection bias associated with the retrospective nature of this study. Therefore, further validation is needed through prospective studies with larger sample sizes conducted across multiple centers in the future. Furthermore, this study only focuses on the clinical

perspective, and future research is needed to further investigate the correlation between bronchiolitis and BO development in terms of pathogenesis from the perspectives of immunology and molecular biology.

In summary, MP bronchiolitis patients with a history of allergic diseases, wheezing sounds, and involvement of at least three lung lobes are prone to developing hypoxemia. And those who experience hypoxemia recover more slowly during short-term follow-up and have a higher incidence of BO.

## Supporting information

**S1 Table: Clinical data, laboratory indicators and imaging data of children with BO sequelae.**
(DOCX)

## Acknowledgments

The authors are grateful to the parents and children for their participation and multiple visits throughout the study.

## Author contributions

**Conceptualization:** Yu Chen, Chenxi Lin.

**Data curation:** Min Zhang, XingQian Lai, QiaoRu Lin.

**Formal analysis:** Rui Huang, Qi Chen.

**Investigation:** Yu Chen, Min Zhang, XingQian Lai, QiaoRu Lin.

**Methodology:** Yu Chen, Chenxi Lin, Rui Huang.

**Project administration:** Yu Chen, Chenxi Lin, Ling Chen.

**Supervision:** Yu Chen, Chenxi Lin, Ling Chen.

**Validation:** Yu Chen, Chenxi Lin, Rui Huang, Qi Chen, XingQian Lai, Ling Chen.

**Visualization:** Rui Huang, Qi Chen, Min Zhang, QiaoRu Lin.

**Writing – original draft:** Yu Chen, Chenxi Lin.

**Writing – review & editing:** Yu Chen, Chenxi Lin.

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
