## [Decision Letter · Decision Letter 0]

17 Sep 2025

Dear Dr. Chen,

We look forward to receiving your revised manuscript.

Kind regards,

Oliver Schildgen

Academic Editor

PLOS ONE

Journal Requirements:

Additional Editor Comments:

**Please address all reviewers' comments when you prepare your revised manuscript. **

Reviewer's Responses to Questions

**Comments to the Author**

1. Is the manuscript technically sound, and do the data support the conclusions?

Reviewer #1: Yes

Reviewer #2: Yes

2. Has the statistical analysis been performed appropriately and rigorously?

Reviewer #1: Yes

Reviewer #2: Yes

3. Have the authors made all data underlying the findings in their manuscript fully available?

Reviewer #1: Yes

Reviewer #2: No

4. Is the manuscript presented in an intelligible fashion and written in standard English?

Reviewer #1: Yes

Reviewer #2: Yes

Reviewer #1: The article is concise and well-structured, but the writing in the discussion section needs improvement.

General comments

The article states, "The data were anonymously provided by the Information Department, and the collection was completed on May 14, 2025. Due to its retrospective nature and anonymous processing, informed consent was waived".

Did the ethics committee authorize the clinical study given the lack of informed consent?

In the statistical analysis section, there is a lack of description of the goodness-of-fit metrics used for the binary logistic regression analysis and the ROC curve that was utilized.

Discussion Section

In the comparison of clinical data, you could analyze bacterial load to determine if it is a direct risk factor for the progression of the infection to a greater number of lobes or if it is associated with the persistence of inflammation in the alveolar tissue.

In Figure 2, a comparison image showing a healthy lung would be helpful.

A better explanation is needed for why children with allergies are more vulnerable to an exaggerated response when they are infected by Mycoplasma pneumoniae (MP), from an immunological perspective and considering recent research.

It is mentioned that from an immunological perspective, macrophages are responsible for bronchiolitis, however, there is a cellular response where other cell types have a greater impact on tissue damage. You also failed to mention the damage caused by CARDS toxins.

Reviewer #2: The manuscript presents a retrospective review of cases of Mycoplasma pneumoniae induced bronchiolitis in children treated at a single centre.

Comments:-

Methods - diagnosis of M. pneumoniae was based on either serology or nucleic acid testing. What proportion were diagnosed by serology alone - how was acute serology differentiated from prior exposure. Was convalescent serology performed to confirm a rising titre. Was there a difference between children diagnosed by serology v's nucleic acids testing?

Results - Univariate analysis suggests a correlation of hypoxaemia with history of allergic diseases, wheezing sounds, and the number of infected lung lobes. It is likely that these factors are not independent of one another. Completion of multivariate analysis should be considered?

**Do you want your identity to be public for this peer review?** For information about this choice, including consent withdrawal, please see our Privacy Policy

Reviewer #1: No

Reviewer #2: No

---

## [Author Response · Author response to Decision Letter 1]

12 Oct 2025

Dear Editor Oliver Schildgen and Reviewers,

I am writing to thank you and the esteemed reviewers for your thoughtful and constructive comments on our manuscript. We have carefully reviewed each of the points raised by the reviewers, and we appreciate the time and effort invested in providing valuable feedback that has significantly improved the quality and clarity of our work. Below, we provide a point-by-point response to the reviewers' comments:

Reviewer #1

Comment 1: The article states, "The data were anonymously provided by the Information Department, and the collection was completed on May 14, 2025. Due to its retrospective nature and anonymous processing, informed consent was waived".Did the ethics committee authorize the clinical study given the lack of informed consent?

Response: Thank you for raising this important point regarding ethics approval and informed consent. We confirm that our study was conducted in full compliance with the principles of the Declaration of Helsinki and was formally reviewed and approved by the Ethics Committee of Zhongshan Hospital, Xiamen University.

As specified in the “English translation for ethics approval document.docx”, our study application materials included a "Waiver of Informed Consent", which the ethics committee fully discussed during the review process. Additionally, the data used in this research are secondary data anonymously provided by the hospital’s Information Department; all individual identifiers (such as names, medical record numbers) have been completely de-identified, and no identifiable information was accessed or used during the research process. This ensures the study poses minimal risk to participants and will not adversely affect their rights and welfare.

We appreciate the opportunity to clarify this matter and assure you that all ethical standards were strictly followed throughout the study.

Comment 2: In the statistical analysis section, there is a lack of description of the goodness-of-fit metrics used for the binary logistic regression analysis and the ROC curve that was utilized.

Response: We agree that providing a detailed description of the statistical methods is essential for methodological integrity. To address this, we have thoroughly revised the “Statistical Analysis” section (lines 144-146) to include the missing details:

multivariate binary logistic regression was performed to determine the independent risk factors associated with hypoxemia, and the model's goodness-of-fit was assessed using the Hosmer-Lemeshow test. The predictive performance of significant continuous variables was evaluated using receiver operating characteristic (ROC) curves. The area under the curve and its 95% confidence interval were calculated. The optimal cut-off value, sensitivity, and specificity were determined based on the maximum Youden's index.

Additionally, we have added the sentence " The Hosmer-Lemeshow test showed a P-value of 0.315, suggesting good goodness-of-fit of the model." at line 208 in the section "Analysis of risk factors for hypoxemia in MP bronchiolitis".

We believe these revisions have significantly improved the clarity and completeness of our statistical reporting.

Comment 3: In the comparison of clinical data, you could analyze bacterial load to determine if it is a direct risk factor for the progression of the infection to a greater number of lobes or if it is associated with the persistence of inflammation in the alveolar tissue.

Response: Thank you very much for your valuable comment. We fully acknowledge the importance of bacterial load, which supplements a crucial potential analytical dimension to our study and helps refine the research conclusions.

Fortunately, all patients underwent MP-DNA load testing via secretions, and we collected these data through the Information Department. There were a total of 14 MP-DNA-negative cases which were diagnosed by paired serum antibody titer changes, with 4 cases in the hypoxemia group and 10 in the control group. After excluding these negative cases, Spearman correlation analysis showed no significant association between MP DNA load and the number of infected lung lobes (correlation coefficient = -0.052, P = 0.512), indicating MP DNA load may not be a direct risk factor for increased infected lung lobes.

Given our current limited understanding, assessing alveolar inflammation persistence is challenging, a difficulty further compounded by the lack of direct biochemical indicators. Given most patients had no dynamic follow-up of indicators like CRP or lung CT, we used symptom/sign recovery time and lung function recovery time as proxy measures. After excluding MP-DNA-negative cases and BO cases (recovery time unquantifiable), Spearman analysis revealed no correlation between MP-DNA load and either recovery time (symptoms/signs: correlation coefficient = -0.051, P = 0.524; lung function: correlation coefficient = -0.013, P = 0.870).

To reflect these new conclusions, the following revisions have been made:

1. "Statistical analysis" section (Line 144): Spearman correlation analysis was adopted to assess the correlation between relevant variables.

2. Add a column of "MP-DNA load (copy/ml)" to Table 1, and the comparison of P-value between the two groups is 0.337.

3. "Comparison of imaging data" section (Line 195): After excluding 4 MP-DNA-negative cases in the hypoxemia group and 10 MP-DNA-negative cases in the control group, no correlation was found between MP-DNA load and the number of infected lung lobes (correlation coefficient = -0.052, P = 0.512)

4. "Results of follow-up" section (Line 227): After excluding MP-DNA-negative cases and BO cases, MP-DNA load showed no correlation with the recovery time of symptoms and signs (correlation coefficient = -0.051, P = 0.524) or the recovery time of lung function (correlation coefficient = -0.013, P = 0.870).

Comment 4: In Figure 2, a comparison image showing a healthy lung would be helpful.

Response: Thank you for your valuable suggestion. We have added a comparison image of a healthy lung (Image J) to enhance the visual comparison.

Comment 5: A better explanation is needed for why children with allergies are more vulnerable to an exaggerated response when they are infected by Mycoplasma pneumoniae (MP), from an immunological perspective and considering recent research. It is mentioned that from an immunological perspective, macrophages are responsible for bronchiolitis, however, there is a cellular response where other cell types have a greater impact on tissue damage. You also failed to mention the damage caused by CARDS toxins.

Response: We highly appreciate your constructive suggestions, which have guided us to deepen our understanding of the pathological mechanisms underlying MP infection and BO formation. Through a review of relevant literature, we have further clarified the roles of neutrophils, T lymphocytes, and the CARDS toxin in these processes. We have therefore carefully revised the relevant sections of the Discussion as follows:

1. The second paragraph of Discussion (Lines 237-249) has been revised to “Atopic constitutions usually manifest as a history of allergic diseases and positive allergen test results [14]. A few studies have suggested that children with an atopic constitution are more susceptible to bronchiolitis [15], suggesting the importance of considering atopic constitution in predicting the severity of MP bronchiolitis. Dysregulation and overactivation of the immune response are observed following MP infection. The MP-secreted Community-Acquired Respiratory Distress Syndrome (CARDS) toxin facilitates upregulation of cytokines interleukin-4 and interleukin-13, thereby amplifying eosinophilic airway inflammation and promoting a T-helper 2 (Th2) cell response [16]. The pre-existing Th2 - predominant state in individuals with atopic constitutions overlaps with the effects of the CARDS toxin, leading to an amplified and prolonged eosinophilic inflammatory response. Additionally, individuals with an atopic constitution may have a higher MP load in their airways [14, 17], accompanied by delayed pathogen clearance, which in turn increases the risk of severe disease [18, 19]. Our study indicates that a history of allergic diseases is an independent risk factor for hypoxemia. Therefore, assessing patients' allergic status during the initial evaluation is crucial for the timely identification of early severe MP bronchiolitis.”

2. The fifth paragraph of Discussion (Lines 267-277) has been revised to “One hypothesis is that diffuse bronchiolar lesions and the occurrence of BO appear to be related, as suggested by Huang et al. and Zhao et al. [1,4]. In this study, the hypoxemia group had involvement of at least three lung lobes, correlating with an elevated risk of developing BO. Hence, our research supports the hypothesis from a clinical observation perspective. The pathogenesis of BO is a continuous process [21]. During MP infection, the CARDS toxin directly induces necrosis and shedding of airway epithelial cells [16]. Meanwhile, infiltrated neutrophils release proteolytic enzymes and reactive oxygen species, contributing to airway damage [22]. Additionally, CD8+ cytotoxic T lymphocytes target infected host cells, amplifying epithelial injury. These factors collectively exacerbate airway epithelial destruction and exfoliation [23]. Exposure of the damaged basement membrane promotes the formation of reticular exudates by fibrinogen in the bronchiolar lumen, which further initiates the organization process, induces fibroblast infiltration, proliferation, and collagen deposition, and ultimately leads to the formation of irreversible scar tissue, resulting in airway obstruction. Macrophages may also serve a bridging role in this pathological process by regulating inflammatory and fibrotic responses [24, 25].”

We hope these revisions enhance the manuscript’s mechanistic discussions. Thank you again for your insights.

Reviewer #2

Comment 1: Methods - diagnosis of M. pneumoniae was based on either serology or nucleic acid testing. What proportion were diagnosed by serology alone - how was acute serology differentiated from prior exposure. Was convalescent serology performed to confirm a rising titre. Was there a difference between children diagnosed by serology v's nucleic acids testing?

Response: Thank you for your valuable comments on our manuscript, and we appreciate your attention to study details and are pleased to provide supplementary information on MP infection diagnosis as requested.

From January 2017 to December 2024, 4,538 children in our hospital were diagnosed with MP infection, of whom approximately 90.8% tested positive for MP DNA. Repeated MP DNA testing was done for patients with negative initial results but poor treatment response and MP-infection-suggestive symptoms/imaging to reduce false negatives; MP DNA with <10³ copy/ml was excluded to rule out residual nucleic acids from prior infections. Patients with negative MP DNA results were diagnosed through serological testing.

For MP serology, paired serum comparison was used, with a 4-fold antibody titer change indicating current infection. We recommended all patients complete paired serum testing to confirm MP infection; however, some parents refused convalescent serum recheck due to personal reasons. After re-reviewing the data, we found that 2,313 cases (51.0%) had positive paired sera; 879 cases (19.4%) had negative initial sera or positive initial sera without 4-fold convalescent titer change; 1,346 cases (30.0%) had positive initial sera but refused convalescent recheck. Patients whose serological results did not meet the diagnostic criteria all had positive MP DNA test results.

Among the 178 selected MP bronchiolitis cases: 164 (92.1%) were MP DNA-positive; 113 (63.5%) had positive paired sera; 36 (20.2%) had negative serology (negative initial sera or positive initial sera without 4-fold convalescent titer change); 29 (16.3%) refused serological recheck.

To clarify, we made the following revisions:

1. In the "Patients" section (Lines 79-80), we revised the content to: "among which 4,538 cases were confirmed to have MP infection through paired serum antibody testing and/or nucleic acid testing of secretions"

2. In the "Comparison of Demographic and Clinical Data" section (Line 156), we added:"Of these, 164 cases (92.1%) were positive for MP DNA, while 113 cases (63.5%) had positive paired serum tests, 36 cases (20.2%) showed negative serological test results (negative initial sera or positive initial sera without 4-fold convalescent titer change), and 29 cases (16.3%) did not undergo a recheck of serum antibodies."

According to our statistics, there were no significant differences in clinical symptoms, signs, laboratory indicators, or imaging between serology diagnosed cases and nucleic acid diagnosed cases.

We hope this supplementary information and manuscript revision address your concerns.

Comment 2: Results - Univariate analysis suggests a correlation of hypoxaemia with history of allergic diseases, wheezing sounds, and the number of infected lung lobes. It is likely that these factors are not independent of one another. Completion of multivariate analysis should be considered?

Response: We would like to express our gratitude for your thoughtful comment concerning the potential interdependence of factors associated with hypoxemia and your recommendation to perform a multivariate analysis.

To address your concern, we have expanded the variable set and implemented a stepwise analytical approach. Specifically, we first conducted univariate logistic regression to screen potential risk factors, followed by collinearity testing to assess variable independence. Subsequently, multivariate logistic regression was performed to identify independent predictors. The revised content in the "Analysis of risk factors for hypoxemia in MP bronchiolitis" section (Lines 204-210) is as follows:

“The presence or absence of hypoxemia was used as the dichotomous dependent variable, while the independent variables included a history of allergic diseases, positive aeroallergen test, positive food allergen test, wheezing sounds, CRP, D-dimer, and the number of infected lung lobes. Univariate binary logistic analysis showed that only D-dimer was excluded due to its P-value > 0.2, whereas all other variables with a P-value < 0.2 were carried forward to subsequent analyses. Collinearity analysis based on linear regression was conducted after the two continuous variables, CRP and the number of infected lung lobes, had been subjected to Z-score standardization. The analysis revealed that the variance inflation factors of these variables were 1.191, 1.211, 1.032, 1.099, 1.302, and 1.348, respectively, indicating no significant multicollinearity. Thus, they were included in the multivariate binary logistic regression model. The results demonstrated that a history of allergic diseases, wheezing sounds, and the number of infected lung lobes were independent risk factors for hypoxemia (Fig 3).”

Correspondingly, we revised the sentence in the "Statistical analysis" section (Lines 144-145) to the following sentence:"Variables with a P-value < 0.2 were sequentially subjected to univariate binary logistic regression analysis for variable selection and linear regression for collinearity exclusion, after which multivariate binary logistic regression was performed to determine the independent risk factors associated with hypoxemia."

Once again, we would like to extend our heartfelt gratitude to the editor for affording us the opportunity to revise our manuscript, and to the reviewers who have generously dedicated their time to scrutinizing our work. We sincerely hope that the revisions incorporated into the manuscript, along with our accompanying responses, will be deemed adequate to make our work worthy of publication.

Yours sincerely,

Yu Chen

Department of Pediatrics, Zhongshan Hospital, Xiamen University, China

Email: hhhanleng@xmu.edu.cn

---

## [Editor Report · Decision Letter 1]

14 Oct 2025

Mycoplasma Pneumoniae Bronchiolitis and Hypoxemia: A Retrospective Cohort Study on Risk and Prognosis

PONE-D-25-25034R1

Dear Dr. Chen,

We’re pleased to inform you that your manuscript has been judged scientifically suitable for publication and will be formally accepted for publication once it meets all outstanding technical requirements.

Kind regards,

Oliver Schildgen

Academic Editor

PLOS ONE
---

## [Editor Report · Acceptance letter]

PONE-D-25-25034R1

PLOS ONE

Dear Dr. Chen,

I'm pleased to inform you that your manuscript has been deemed suitable for publication in PLOS ONE. Congratulations! Your manuscript is now being handed over to our production team.

Kind regards,

on behalf of

Professor Oliver Schildgen

Academic Editor

PLOS ONE